**Subject Area:**
psychology

restrained eating, dietary restraint, assessment measures, food craving, disinhibited eating, body mass index

**Author for correspondence:**
Rachel C. Adams
e-mail: adamsrc1@cardiff.ac.uk

# Do restrained eaters show increased BMI, food craving and disinhibited eating? A comparison of the Restraint Scale and the Restrained Eating scale of the Dutch Eating Behaviour Questionnaire

Rachel C. Adams[1], Christopher D. Chambers[1]
and Natalia S. Lawrence[2]

[1]CUBRIC, School of Psychology, Cardiff University, Maindy Road, Cardiff CF24 4HQ, UK
[2]School of Psychology, College of Life and Environmental Sciences, University of Exeter, Exeter EX4 4QG, UK

RCA, 0000-0002-8053-0671; CDC, 0000-0001-6058-4114; NSL, 0000-0003-1969-6637

Despite being used interchangeably, different measures of restrained eating have been associated with different dietary behaviours. These differences have impeded replicability across the restraint literature and have made it difficult for researchers to interpret results and use the most appropriate measure for their research. Across a total sample of 1731 participants, this study compared the Restraint Scale (RS), and its subscales, to the Dutch Eating Behaviour Questionnaire (DEBQ) across several traits related to overeating. The aim was to explore potential differences between these two questionnaires so that we could help to identify the most suitable measure as a prescreening tool for eating-related interventions. Results revealed that although the two measures are highly correlated with one another ($r$s = 0.73–0.79), the RS was more strongly associated with external ($r$s = −0.07 to 0.11 versus −0.18 to −0.01) and disinhibited eating ($r$s = 0.46 versus 0.31), food craving ($r$s = 0.12–0.27 versus 0.02–0.13 and 0.22 versus −0.06) and body mass index ($r$s = 0.25–0.34 versus −0.13 to 0.15).

The results suggest that, compared to the DEBQ, the RS is a more appropriate measure for identifying individuals who struggle the most to control their food intake.

## 1. Introduction

Dietary restraint refers to the tendency to chronically limit food intake in order to lose weight or prevent weight gain. In today's 'obesogenic' environment such restrained eating appears to be an adaptive behaviour. Paradoxically, however, high dietary restraint has been associated with increased impulsivity [1–3], heightened reactivity towards food [4–8] and disinhibited eating [9–12]. It is unclear whether dietary restraint is a cause or effect of impulsive or disinhibited eating [13], resulting in conflicting advice to would-be dieters, and uncertainty among researchers about what dietary restraint scales actually measure. To clarify this issue, the present study examined associations between common measures of dietary restraint and traits associated with overeating, including food craving, disinhibited eating and body mass index (BMI).

The association between dietary restraint and overeating has led to recommendations to relax restraint in order to promote healthy eating [14]; however, this has proved controversial due to inconsistencies across findings. Restrained eating is typically associated with 'counterregulation' of food intake whereby an individual will consume more calories following a small amount of palatable food (known as a preload), compared to a non-restrained eater who will consume fewer calories [10,15]. However, not all researchers have been able to replicate these effects [16,17], and others have argued that the opposing causal relationship could exist whereby restrained eating is a response to weight gain [13,18–20] and may even have a protective role [21,22]. For example, in a sample of adults with morbid obesity, Brogan & Hevey [23] found that restrained eating was negatively associated with total food intake and the consumption of both high-fat and high-sugar foods. Other researchers have proposed that certain types of restraint (i.e. flexible versus rigid) are adaptive and protect against weight gain [24].

It is thought that the conflicting findings in the restraint literature are most likely due to the use of different measures of dietary restraint [13,25]. The original psychometric tool that showed support for the counterregulation of food intake in the formulation of restraint theory is the Restraint Scale (RS) [10,26]. However, the RS has been criticized for criterion confounding and has been shown to consist of two distinct subscales: the concern for dieting scale and the weight fluctuation scale [27–31]. The concern for dieting subscale is believed to reflect a greater attentional and emotional association with food (i.e. feeling conscious of one's food intake and feeling guilt after overeating), whereas the latter scale assesses weight history and weight fluctuations. It is this weight fluctuation scale that may be responsible for associations between restraint and overeating due to the measurement of absolute weight changes and increased attempts to compensate for weight gain [12,28,31–33]. On the other hand, the concern for dieting scale has also shown strong correlations with self-reported measures of binge eating and dietary disinhibition [25,34]. This relationship may reflect further criterion confounding as some researchers have argued that certain items (e.g. 'Do you eat sensibly in front of others and splurge alone?') directly measure disinhibited or opportunistic eating [12,17,35].

To overcome the issues of criterion confounding in the RS, other measures of restrained eating such as those in the Dutch Eating Behaviour Questionnaire (DEBQ; [36]) and the Three Factor Eating Questionnaire (TFEQ; [12]) were developed. Contrary to the RS, the restraint scales in these measures have been shown to be associated with reduced calorie intake and successful dieting [17,23,34,37] and are not associated with the effect of preloading on counterregulation of intake [16,17]. Therefore, despite findings that these three measures of dietary restraint are highly correlated with one another [25,34,38,39], there are important differences between these measures that should be acknowledged (for reviews, see [13,29,40,41]). This may be especially true when preselecting individuals for dietary interventions based on measures of restraint.

One particular intervention that has targeted restrained eaters is food-related self-control training [42–45]. In this intervention, individuals who are trained to stop, or inhibit, their responses to unhealthy palatable foods consume fewer calories than those who perform an active control task. Importantly, Houben & Jansen [43], Lawrence et al. ([44]; Study 2) and Veling et al. [45] have all shown a moderating effect of dietary restraint, whereby restrained eaters benefit more from self-control training compared to unrestrained eaters. However, each of these studies used a different measure of dietary restraint. Houben & Jansen [43] used the original RS [26], Veling et al. [45] used

**Table 1.** Measures recorded for each of the four samples. Note: sample 4 is a subset of sample 1; all participants in sample 4 were eligible for another research study and had RS scores of 15+. The only measures analysed for this sample were the experimenter-recorded BMI and G-FCQ-T. R, Recorded but not analysed; RS, Restraint Scale; DEBQRE, Dutch Eating Behaviour Questionnaire Restrained Eating scale; DEBQEE, Dutch Eating Behaviour Questionnaire External Eating scale; ACQC, Attitudes to Chocolate Questionnaire—Craving scale; TFEQD, Three Factor Eating Questionnaire Disinhibition scale; BMI, body mass index; G-FCQ-T, General Food Craving Questionnaire—Trait version.

| | sample 1 N = 1320 | sample 2 N = 207 | sample 3 N = 202 | sample 4 N = 245 |
|---|---|---|---|---|
| RS | ✓ | ✓ | ✓ | R |
| DEBQRE | ✓ | ✓ | ✓ | R |
| DEBQEE | ✓ | ✓ | | R |
| ACQC | ✓ | ✓ | | R |
| TFEQD | | | ✓ | |
| BMI (self-report) | | | ✓ | |
| BMI (objective) | | | | ✓ |
| G-FCQ-T | | | | ✓ |

only the concern for dieting subscale of this measure (RSCD), and Lawrence *et al.* [44] used the restrained eating subscale of the DEBQ (DEBQRE; [36]). At a time when dietary restraint is being considered as a major moderator for food-related interventions, we believed it was important to investigate these measures using a large and current sample.

In the current study, we explored potential differences between the RS, the RSCD [26] and the DEBQRE [36] in a sample of 1731 participants. For completeness, we have also included the weight fluctuation scale of the RS (RSWF). We first examined the internal consistency, factor structure and demographic differences for these measures of restrained eating to see whether they were in accordance with previous research [30,34,38]. We then explored correlations within different measures of restraint and between these measures and external eating, disinhibited eating, food craving and BMI. The aim of this study was to explore potential differences across restraint measures so that we could help to identify the most appropriate prescreening tool for eating-related interventions. In accordance with previous research, we expected to find that the RS was more strongly associated with disinhibited eating and BMI compared to the DEBQRE.

## 2. Method

### 2.1. Participants

There were four samples included in this study. Sample 1 ($N = 1320$) was mainly staff and students from Cardiff University who responded to advertisements for a study investigating food and positive emotion (1031 females; age range: 17–66, $M = 22.2$, s.e. = 0.19). Samples 2 ($N = 207$) and 3 ($N = 202$) were undergraduate psychology students (sample 1: 185 females; age range: 17–42, $M = 18.61$, s.e. = 0.17; sample 2: 178 females; age range: 17–50, $M = 19.17$, s.e. = 0.25). Sample 4 ($N = 245$; 231 females; age range: 18–61, $M = 22.26$, s.e. = 0.46) was a subset of sample 1 who scored highly on the RS (15+; 108 participants also scored highly [10+] on the ACQC). Table 1 shows an overview of measures recorded for each sample. All methods were approved by the School of Psychology Research Ethics Committee, Cardiff University.

### 2.2. Measures and materials

#### 2.2.1. The Restraint Scale

The RS [26] is a 10-item questionnaire and total scores range from 0 to 35. A total score of 15+ has previously been used as a cut-off to indicate 'restrained eating' (e.g. [43,46,47]). There are considered to be two subscales of the RS: concern for dieting (RSCD) and weight fluctuations (RSWF; [27–31]).

The RSCD subscale includes six questions regarding dieting frequency and feelings towards weight gain and overeating (for example, 'Do you have feelings of guilt after overeating?'; range 0–19); the RSWF subscale has four questions regarding weight loss and weight gain (for example 'What is the maximum amount of weight (in pounds) that you have ever lost within one month?'; range 0–16). Internal consistency for the RS has been shown to be in the acceptable–good range ($\alpha$ =0.78–0.86; RSCD = 0.78–0.79; RSWF; 0.69–0.72; [27,38]) and strong test–retest reliability has been reported ($r$ = 0.91–0.95; [38]; for discussions on validity, see [27,38]).

### 2.2.2. The Dutch Eating Behaviour Questionnaire—Restrained Eating scale

The DEBQ [36] is a 33-item questionnaire measuring restrained, emotional and external eating behaviour. The restrained eating scale (DEBQRE) includes 10 questions regarding restriction or avoidance of food intake (range 10–50). For example, 'If you have put on weight, do you eat less than you usually do?' Internal consistency is reportedly good–excellent ($\alpha$ = 0.89–0.95) and test–retest reliability is strong ($r$ = 0.92; [34,38]).

### 2.2.3. The Dutch Eating Behaviour Questionnaire—External Eating scale

The external eating subscale of the DEBQ (DEBQEE) includes 10 questions concerning eating, and overeating, as a result of external food cues such as the taste, sight and smell of food, as well as overeating as the result of seeing others eating (range 10–50). For example, 'If food smells and looks good to you, do you eat more than usual?' Internal consistency is good ($\alpha$ = 0.80–0.81; [36,37,48]).

### 2.2.4. The Attitudes to Chocolate Questionnaire—Craving scale

The chocolate craving subscale of the ACQ (ACQC; [49]) has 10 items that concern wanting and desire for chocolate as well as a lack of control over chocolate consumption (range −30 to 30). The questions concern wanting and desire for chocolate (for example, 'My desire for chocolate often seems overpowering') as well as a lack of control over chocolate consumption (for example, 'Even when I do not really want any more chocolate I will often carry on eating it'). Internal consistency is good–excellent ($\alpha$ = 0.80–0.81) and test–restest reliability is moderate ($r$ = 0.69; [48,50]).

### 2.2.5. The Three Factor Eating Questionnaire—Disinhibition scale

The disinhibition subscale of the TFEQ (TFEQD; [12]) includes 16 questions related to overeating. The first question was re-worded from the original 'When I smell a sizzling steak or see a juicy piece of meat, I find it very difficult to keep from eating, even if I have just finished a meal' to consider vegetarians and those who do not consider meat to be the most desired food [51]. In keeping with recommendations from revised versions of this questionnaire, we also coded responses to the first 13 questions on a four-point scale from 1, 'Definitely false' to 4, 'Definitely true' [51,52]. Questions 14–16 were coded on the original 4 point scales (range 16–64). Note that question 14 is identical to question 6 from the RSCD subscale, it was therefore removed when comparing correlations between the TFEQD and the RS measures. Internal consistency is excellent ($\alpha$ = 0.91; [12]).

### 2.2.6. The General Food Craving Questionnaire—Trait version

The G-FCQ-T [53,54] is a 21-item questionnaire measuring the strength of food cravings (range 21–126). There are four craving subscales including: preoccupation with food (for example, 'I feel like I have food on my mind all the time'; six questions; range 6–36), loss of control over food intake (for example, 'If I eat what I'm craving, I often lose control and eat too much'; six questions, range 6–36), positive outcome expectancy (for example, 'Eating what I crave makes me feel better'; five questions, range 5–30) and emotional craving (for example, 'I crave foods when I'm upset'; four questions, range 4–24). Internal consistency is reported to be excellent ($\alpha$ = 0.94–0.97) and test–retest reliability as strong ($r$ = 0.88; [53,54]).

### 2.2.7. Body mass index

Participants in sample 3 self-reported their height and weight and those in sample 4 (who participated in the experimental studies) had their height and weight recorded by the experimenter using a tape measure and mechanical bathroom scales. These values were used to calculate BMI ($kg\ m^{-2}$).

## 2.3. Procedure

Participants in samples 1 and 2 received the questionnaires in the same order; the DEBQRE and DEBQEE were followed by the RS and the ACQC. Sample 3 received the DEBQRE followed by the RS, the TFEQD and measures of height and weight. All participants completed the questionnaires remotely. Those in samples 2 and 3 completed the questionnaires via an Internet survey, whereas those in sample 1 could choose to answer the questionnaires electronically via email, in hard copy or via an Internet survey. All participants in sample 4 were recruited for a behavioural study that included the G-FCQ-T (the full details can be found in [42]). The height and weight of these participants was recorded at the end of the study to calculate BMI.

## 2.4. Statistical analysis

### 2.4.1. Internal consistency and factor analysis

Internal consistency and factor structures of the questionnaire measures were explored to ensure that the data were consistent with previous findings [30,34,38]. These analyses were performed on samples 1–3 separately to consider the replicability of these findings and on the total sample for completeness. Internal consistency was calculated using Cronbach's alpha. The factor structure of the RS and DEBQRE were explored using principal components analysis with varimax rotation in accordance with previous research [27,30,34,38]. Factors were extracted based on having eigenvalues greater than 1.

### 2.4.2. Demographic differences

Demographic differences for age and gender in restrained eating were explored; for this analysis, data were collapsed across samples 1–3 to increase overall heterogeneity (due to the homogeneity in the undergraduate samples, i.e. samples 2 and 3).

### 2.4.3. Comparison of the RS and DEBQRE

Similarities and differences between the RS, RSCD, RSWF and DEBQRE were then explored with inter-correlations between these scales as well as correlations with external eating (DEBQEE), food-related disinhibition (TFEQD), food craving (ACQC and G-FCQ-T) and BMI.

### 2.4.4. Statistical power and procedures

Correlations for sample 1 were well powered; the minimum sample size across all comparisons was 1306. A sensitivity analysis revealed that the smallest detectable effect size with 90% power and $\alpha = 0.05$ was $r = 0.09$ (using G*Power; [55]). Samples 2, 3 and 4 had smaller sample sizes with a minimum $N$ of 207, 202 and 213, respectively, across all comparisons; these sample sizes enabled detection of $r \geq 0.22$ with 90% power. To account for missing data across questionnaires, all correlations use individual mean scores rather than the total scores. Cases were removed from analyses where no data were available. All results are reported with unadjusted significance values; corrections for multiple comparisons were calculated for all within-test, within-sample analyses and are only reported where these corrections changed the interpretation of an analysis from statistically significant to not statistically significant. Differences between significant correlations were also analysed using Fisher's $Z$-tests and Steiger's $Z$-tests using online resources (http://vassarstats.net/rdiff.html; http://www.psychmike.com/dependent_correlations.php). All other analyses were carried out using SPSS. All study data and analysis scripts are available online (https://osf.io/gsfrj/).

# 3. Results

A summary of all descriptive statistics can be found in table 2.

## 3.1. Internal consistency

Internal consistency was high ($\alpha > 0.8$) for the RS, DEBQRE, DEBQEE, ACQC and TFEQD and was satisfactory ($\alpha > 0.7$) for RSCD and RSWF (see the electronic supplementary material, table S.1).

**Table 2.** Descriptive statistics across all samples (s.e. within parentheses). Note: RS, Restraint Scale; RSCD, concern for dieting subscale of the RS; RSWF, weight fluctuation scale of the RS; DEBQRE, Dutch Eating Behaviour Questionnaire Restrained Eating scale; DEBQEE, Dutch Eating Behaviour Questionnaire External Eating scale; TFEQD, Three Factor Eating Questionnaire Disinhibition scale; ACQC, Attitudes to Chocolate Questionnaire—Craving scale; G-FCQ-T, General Food Craving Questionnaire—Trait Version; BMI, body mass index.

| | sample 1 (min $N = 1232$) | sample 2 (min $N = 207$) | sample 3 (min $N = 195$) | sample 4 (min $N = 222$) |
|---|---|---|---|---|
| age | 22.2 (0.19) | 18.61 (0.17) | 19.17 (0.25) | |
| sex (% female) | 80 ($n = 1031$) | 89.4 ($n = 185$) | 88.1 ($n = 178$) | |
| RS | 13.17 (0.16) | 12.59 (0.45) | 12.45 (0.43) | |
| RSCD | 8.47 (0.1) | 7.66 (0.28) | 8.15 (0.29) | |
| RSWF | 4.75 (0.09) | 4.94 (0.22) | 4.3 (0.21) | |
| DEBQRE | 25.98 (0.23) | 26.43 (0.67) | 27.67 (0.64) | |
| DEBQEE | 34.02 (0.17) | 32.91 (0.45) | | |
| TFEQD | | | 38.69 (0.53) | |
| ACQC | 0.55 (0.42) | −7.08 (0.94) | | |
| G-FCQ-T total | | | | 3.46 (0.07)[a] |
| BMI (kg m$^{-2}$)[b] | | | 22.11 (0.26) | 24.7 (0.29) |

[a]Only mean scores were collected for G-FCQ-T.
[b]BMI was self-reported for sample 3 and was objectively measured by a researcher for sample 4.

## 3.2. Factor structure

Extraction for the RS revealed two factors consistent with the RSCD and RSWF subscales and explained a similar amount of variance (see the electronic supplementary material, tables S.2 and S.3; [27–30,38]). Consistent with previous research, the DEBQRE revealed just one factor [36,38].

## 3.3. Demographic differences

Consistent with previous findings, we found that females scored significantly higher than males across all restraint measures (all $ps < 0.001$; all $d > 0.39$; see the electronic supplementary material, table S.4; [28,38,39,56,57]). For males, age was positively associated with all measures of restraint (all $rs > 0.17$, all $ps < 0.003$), whereas only the associations with RS and RSWF were significant for females (both $rs > 0.08$, both $ps < 0.004$). These results suggest that women maintain a fairly high level of dietary restraint across time, whereas restraint scores tend to increase with age in men (range reported in this sample: women 17–61; $M = 21.02$; s.e. = 0.15; men 18–66; $M = 23.09$; s.e. = 0.44).

## 3.4. Comparison of the RS and DEBQRE

### 3.4.1. Restraint correlations

Significant positive correlations were found between DEBQRE and all three RS measures for all samples (table 3). For the overall RS, there was a positive correlation with a large effect size across all samples (all $rs > 0.73$, all $ps < 0.001$). This relationship appears to rely to a greater extent on the correlation between DEBQRE and RSCD, which yielded significantly greater correlations (all $zs > 7.57$, all $ps < 0.001$) and effect sizes (all $rs > 0.77$) for all samples compared to the correlation between DEBQRE and RSWF (all $rs < 0.53$).

### 3.4.2. External and disinhibited eating

Although the RS and DEBQRE were highly correlated with one another, they were differentially correlated with external eating (table 3). In sample 1, DEBQEE was positively correlated with RS, RSCD and RSWF (all $ps < 0.01$, all $rs < 0.12$), but was not significantly correlated with DEBQRE

**Table 3.** All correlations between the four measures of restrained eating and all outcome variables. Note: p-values have not been corrected for multiple comparisons. RS, Restraint Scale; RSCD, concern for dieting subscale of the RS; RSWF, weight fluctuation scale of the RS; DEBQRE, Dutch Eating Behaviour Questionnaire Restrained Eating scale; DEBQEE, Dutch Eating Behaviour Questionnaire External Eating scale; TFEQD, Three Factor Eating Questionnaire Disinhibition scale; ACQC, Attitudes to Chocolate Questionnaire—Craving scale; G-FCQ-T, General Food Craving Questionnaire—Trait version; BMI, body mass index.

|  | sample 1 | sample 2 | sample 3 | sample 4 |
|---|---|---|---|---|
| RS—DEBQRE | 0.73*** | 0.79*** | 0.78*** | |
| RSCD—DEBQRE | 0.77*** | 0.84*** | 0.84*** | |
| RSWF—DEBQRE | 0.44*** | 0.53*** | 0.46*** | |
| RS—DEBQEE | 0.11*** | −0.07 | | |
| RSCD—DEBQEE | 0.11*** | −0.10 | | |
| RSWF—DEBQEE | 0.08** | −0.02 | | |
| DEBQRE—DEBQEE | −0.01 | −0.18** | | |
| RS—TFEQD[a] | | | 0.46*** | |
| RSCD—TFEQD[a] | | | 0.43*** | |
| RSWF—TFEQD | | | 0.38*** | |
| DEBQRE—TFEQD | | | 0.31*** | |
| RS—ACQC | 0.27*** | 0.12[#] | | |
| RSCD—ACQC | 0.25*** | 0.11 | | |
| RSWF—ACQC | 0.21*** | 0.1 | | |
| DEBQRE—ACQC | 0.13*** | 0.02 | | |
| RS—G-FCQ-T total | | | | 0.22** |
| RSCD—G-FCQ-T total | | | | 0.18** |
| RSWF—G-FCQ-T total | | | | 0.07 |
| DEBQRE—G-FCQ-T total | | | | −0.06 |
| RS—BMI[b] | | | 0.34*** | 0.25*** |
| RSCD—BMI[b] | | | 0.25*** | −0.04 |
| RSWF—BMI[b] | | | 0.36*** | 0.34*** |
| DEBQRE—BMI[b] | | | 0.15* | −0.13[#] |

***$p < 0.001$, **$p < 0.01$, *$p < 0.05$, #$p < 0.1$.
[a]Question 14 of the TFEQD was removed for these comparisons due to the presentation of the question in the RSCD subscale of the RS.
[b]BMI was self-reported for sample 3 and was objectively measured by a researcher for sample 4.

($r = −0.01$, $p = 0.79$). In sample 2, DEBQEE did not significantly correlate with any RS measures (all $ps > 0.15$); however, there was a significant negative correlation with DEBQRE ($r = −0.18$, $p = 0.008$). For the total sample, only the correlations with the three RS measures remained statistically significant (all $rs > 0.07$, all $ps < 0.01$). The difference in correlations with DEBQEE between DEBQRE and RS was statistically significant ($z = 6.42$, $p < 0.001$; this was also true for both the RSCD and RSWF subscales: both $zs > 4.08$, both $ps < 0.001$).

Results for disinhibited eating revealed a significant positive relationship for all four measures of restrained eating (all $rs > 0.308$, all $ps < 0.001$; table 3). The correlation between the TFEQD and RS was significantly stronger compared with the correlation with DEBQRE ($z = 3.59$, $p < 0.001$). This was largely due to a stronger correlation with RSCD ($z = 3.45$, $p < 0.001$); the difference in correlations between RSWF and DEBQRE was not statistically significant ($z = 1.11$, $p = 0.27$).

### 3.4.3. Food craving

For sample 1, chocolate craving was significantly and positively correlated with all restraint measures (all $rs > 0.13$, all $ps < 0.001$; table 3). The correlation between ACQC and RS was significantly greater than

that with DEBQRE ($z = 6.97$, $p < 0.001$). Both subscales of the RS also had stronger correlations compared to the DEBQRE (RSCD: $z = 6.88$, $p < 0.001$; RSWF; $z = 2.81$, $p = 0.005$). These relationships were not significant in sample 2 (all $rs < 0.12$, all $ps > 0.08$) and showed the same above pattern of results for the total sample.

In the fourth sample, a measure of general food craving was also recorded (using the G-FCQ-T; [53,54]; for completeness, correlations between all subscales for this questionnaire and the restrained eating measures are provided in the electronic supplementary material, table S.5). For the G-FCQ-T total score, there was a significant positive relationship with the RS and RSCD (both $rs > 0.18$, both $ps < 0.007$; table 3). The correlations with DEBQRE and RSWF did not reach statistical significance (both $ps > 0.27$). The correlation between trait food craving and DEBQRE was significantly weaker compared to both the RS and RSCD correlations (both $zs > 3.81$, both $ps < 0.001$), but was not significantly different from RSWF ($z = 1.29$, $p = 0.2$).

### 3.4.4. Body mass index

Sample 3 self-reported their height and weight. BMI was significantly and positively associated with the three RS measures (all $rs > 0.25$, all $ps < 0.001$; table 3) but the correlation with DEBQRE did not survive multiple comparisons ($r = 0.15$, $p = 0.04$; $\alpha = 0.0125$). The correlation between BMI and DEBQRE was significantly weaker compared to all three RS measures (all $zs > 2.54$, all $ps < 0.01$). The correlation between BMI and RSWF was not significantly different from either the RS ($z = 0.45$, $p = 0.65$) or RSCD ($z = 1.65$, $p = 0.1$) measures.

Sample 4 had their height and weight recorded in the laboratory. Correlations revealed a significant positive relationship between BMI and both the RS and RSWF (both $rs > 0.25$, both $ps < 0.001$); these two correlations did not differ significantly ($z = 1.44$, $p = 0.15$). The correlation between BMI and DEBQRE revealed a trend towards a negative relationship that did not survive correction for multiple comparisons ($r = -0.13$, $p = 0.051$; $\alpha = 0.0125$). The relationship between BMI and RSCD was not statistically significant ($r = -0.04$, $p = 0.57$). The correlation between BMI and DEBQRE was significantly weaker compared with the correlations for both RS and RSWF (both $zs > 4.79$, both $ps < 0.001$) but not RSCD ($z = 1.49$, $p = 0.13$).

## 4. Discussion

In the largest study of its kind, we reveal that although the RS and DEBQRE are highly correlated with one another, there are also significant differences between the two measures. In support of the existing literature, we demonstrate that the RS is more strongly associated with external and disinhibited eating, food craving and BMI, when compared to the DEBQRE [6–8,17,29,34,58–61]. These differences have important implications for the ways in which these questionnaires are used.

When comparing the dietary restraint measures, we found strong positive correlations between the RS and DEBQRE (in particular between the RSCD and DEBQRE); these results are consistent with previous findings and suggest that the RS and DEBQRE measure the same construct to a large extent [34,38,39]. However, correlations with external and disinhibited eating measures revealed that the relationships with these eating traits are significantly stronger for the RS compared to the DEBQRE (and showed a medium–large effect size for the relationship between RS and disinhibited eating). Both external and disinhibited eating reflect opportunistic eating in response to environmental cues or initiated food consumption. External eating has been associated with increased attention towards food and has been shown to mediate the relationship between impulsivity and unhealthy food intake [62–64], while, disinhibited eating has been positively correlated with BMI and weight gain [65–67]. Our study is the first to show such differences between the RS and DEBQRE with regard to external and disinhibited eating scales and offers robust, well-powered findings. Our results suggest, therefore, that use of the RS may be more appropriate for identifying individuals who struggle the most with overeating in response to their environment and subsequent weight gain.

Similarly, we found that the RS was more strongly associated with food craving than the DEBQRE. Strong food cravings are also reflective of individuals who are at increased risk of overweight and obesity as they have been associated with poor dietary success, binge eating and increased BMI [56,68–70]. Although we found a positive association between chocolate craving and all measures of restraint, this association was significantly stronger for the RS compared to the DEBQRE. In addition, for the fourth sample of high RS scorers, the RS, but not the DEBQRE, was significantly and positively associated with

general food craving. These results are consistent with other findings showing a positive relationship between the RS and measures of food craving [69,71] and findings showing a negative or non-significant association between food craving and the DEBQRE [56,72–74]. Finally, we provide evidence supporting the finding that the RS is positively associated with BMI using both self-reported and objectively measured BMI [4,8,75]. Associations with the DEBQRE did not reach statistical significance.

Together these findings provide support for previous suggestions that the RS is associated with unsuccessful restraint, whereas the DEBQRE reflects successful restraint [17,29,34,58–61]. We also demonstrate, for the first time, that the RS is more strongly associated with external eating and chocolate craving than the DEBQRE. Our results serve to further illustrate the differences between the two measures of restrained eating, suggesting that the RS may be a better tool to identify those with a tendency to lose control over their eating behaviour and gain weight, whereas the DEBQRE may be more appropriate for identifying those with good food-related self-control [17,29,34,58–61]. In this study, we also considered the difference between the full RS scale and the RSCD subscale based on other researchers using only the RSCD (e.g. [45,76,77]). We found no differences between these two scales for associations with external eating, disinhibited eating, chocolate craving or general food craving, however, there was a significant difference in their associations with BMI due to the stronger correlation with RS compared to RSCD. It has been argued elsewhere that the association between restrained eating and BMI is due to the RSWF subscale (an idea supported by the relatively stronger association between BMI and RSWF reported here), which could be due to the scale measuring absolute changes in weight gain or because the scale reflects increased attempts to compensate for weight gain in those with a higher BMI [12,28,31–33]. Our findings suggest that the weight fluctuation subscale could be important when considering BMI in restrained eaters and that the full RS may be more sensitive than the RSCD scale alone.

The greatest strength of the current study is the large sample size and sufficient statistical power to detect small differences. This has provided us with robust findings on which to base our conclusions. Nevertheless, future work could benefit from replication and extension—particularly across different samples and measures. With our first sample, we attempted to recruit a heterogenous population, however, these participants were self-selected to the extent that they responded to an advert for a food-related study. Such self-selection may have caused bias in our sample. In addition, samples 2 and 3 were undergraduate psychology students, which again narrows our ability to generalize these results to a wider population [78]. The overrepresentation of females in the current study is an issue for wider generalization but is also consistent with food-related research on the whole. An extension including different measures would also be worthwhile, particularly considering different measures of food craving (including cue-induced craving), food-reactivity and sensitivity to reward [79,80]. It is also important to obtain BMI scores across the whole sample such that one could explore how these relationships change with weight status. All of these questions could help us to further understand the nature of restrained eating and the associated measures.

In conclusion, the results of this study suggest that the RS may be a more suitable measure for determining less adaptive 'restrained' eating (that is associated with disinhibited eating), compared with the DEBQRE. These findings have implications for all researchers who wish to consider restrained eating either as a measure or as a pre-selective tool. As the DEBQRE appears to be associated with successful dietary restraint [17,23,34,37], we argue that it is less likely that targeting certain interventions at these individuals will lead to robust and reliable effects on behaviour. The RS, on the other hand, appears to be a reliable and valid tool for identifying individuals who try to control their food intake but struggle the most with disinhibited eating, food craving and weight gain.

Ethics. The study was approved by the School of Psychology Research Ethics Committee, Cardiff University and informed consent was obtained from all participants.

Data accessibility. All study data and analysis scripts are freely available on the Open Science Framework (https://osf.io/gsfrj/).

Authors' contributions. R.C.A. designed the research, collected the data, analysed and interpreted the data and drafted and finalized the manuscript. N.S.L. and C.D.C. made substantial contributions to the design of the research, analysis and interpretation of the data, manuscript drafts and final approval of the manuscript. All the authors agree to be accountable for all aspects of the work in ensuring that questions related to the accuracy or integrity of any part of the work are appropriately investigated and resolved.

Competing interests. C.D.C. is a member of the Royal Society Open Science editorial board but had no involvement in the peer review process of this submission. The authors declare no other competing interests.

Funding. This research was supported by grants held by C.D.C. from the Biotechnology and Biological Sciences Research Council (BB/K008277/1) and the European Research Council (Consolidator grant no. 647893 CCT).

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
