## [Reviewer comments · Royal Society Open Science]

Review History

RSOS-190174.R0 (Original submission)

Review form: Reviewer 1 (Moritz Herle)

Is the manuscript scientifically sound in its present form?

Yes

Are the interpretations and conclusions justified by the results?

Yes

Is the language acceptable?

Yes

Is it clear how to access all supporting data?

Yes

Do you have any ethical concerns with this paper?

No

Have you any concerns about statistical analyses in this paper?

No

Recommendation?

Accept with minor revision (please list in comments)

Comments to the Author(s)

See in the attached (Appendix A).

Review form: Reviewer 2

Is the manuscript scientifically sound in its present form?

Yes

Are the interpretations and conclusions justified by the results?

Yes

Is the language acceptable?

Yes

Is it clear how to access all supporting data?

Yes

Do you have any ethical concerns with this paper?

No

Have you any concerns about statistical analyses in this paper?

Yes

Recommendation?

Major revision is needed (please make suggestions in comments)

Comments to the Author(s)

The goal of this manuscript was to test the associations among two different and commonly used measures of dietary restraint, the Restraint Scale and the Dutch Eating Behavior Questionnaire Restrained Eating (DEBQ-RE) in four different samples. Further, the authors tested the associations of each of these scales with body mass index, non-homeostatic eating (defined as external and disinhibited eating), and food craving (to chocolate and general foods), and compared the strength of the associations with each of the restraint scales with these external correlates. The authors also tested the factor structure and internal consistency of the Restraint Scale and DEBQ-RE. First, the authors found that the Restraint Scale and DEBQ-RE were significantly correlated. Second, dietary restraint was differentially associated with external and disinhibited eating across samples; however, overall, the Restraint Scale demonstrated larger associations with these non-homeostatic eating processes compared to the DEBQ-RE. In general, the RS demonstrated a stronger positive correlation to food craving and body mass index than

the DEBQ-RE. Finally, the authors found that the DEBQ-RE and Restraint Scale factor structures replicated across samples.

Strengths of the study include a large sample size and the study helps the field move toward better understanding the multi-faceted nature of dietary restraint and differential associations among external correlates. The authors are also very clearly familiar with the dietary restraint literature. However, despite these strengths, I have reservations about the manuscript in its current form. My strongest concerns lie within the Introduction, as I feel that the Introduction lacks focus and could be significantly streamlined and also lacks scientific hypotheses to frame the research study. Other, more minor concerns are detailed in the review.

Overall Comments

Please be sure to use person-first language throughout the manuscript when describing weight status in persons. For example, it would be preferable to use the phrase “individuals with obesity” (vs. obese individuals).

This is very minor, but please spell out the first use of body mass index before using the abbreviation of BMI.

Introduction

The authors clearly are well-versed in the dietary restraint literature and the Introduction contains a lot of wonderful information about dietary restraint. At the same time, the Introduction feels a bit unfocused and quite long. I think the manuscript would be much improved if the Introduction were streamlined.

I also would love for the authors to enumerate hypotheses at the end of the Introduction to help frame the study. Based on the prior research, what did you think you might find?

In the first paragraph, I also disagree with the statement that “Overall, rigid diets are not only ineffective for weight loss...” Prescribed, medically monitored very-low calorie diets are effective in promoting weight loss in the short-term. The authors may want to revise this sentence to say that rigid diets are not effective in the long-term (or something of the like).

Method

I would like to see more evidence of reliability and validity of scale scores presented in the description of each measure. Moreover, I would like for internal consistency values to be separated and specified within the description of each measure.

Please report body mass index descriptive statistics (mean, standard deviation, range) for each sample in the Method section rather than in the Results section. BMI is a standard demographic variable for study such as this.

How were missing data handled? I am sure there were some missing data across the four different samples and large overall sample size.

What were the fit statistics of the reported factor analyses? Did the authors use parallel analysis to determine the number of factors to extract?

Were study measures completed online or in-person?

Results

Results were somewhat overwhelming in the way that they were presented. The authors might consider a more concise presentation of their results.

Discussion

Throughout the Discussion, the authors should tone down their conclusions given that this a cross-sectional study.

Decision letter (RSOS-190174.R0)

01-Apr-2019

Dear Dr Adams,

The editors assigned to your paper ("Do restrained eaters show increased BMI, food craving, and disinhibition?

A comparison of the Restraint Scale and the DEBQ Restrained Eating scale") have now received comments from reviewers. We would like you to revise your paper in accordance with the referee and Associate Editor suggestions which can be found below (not including confidential reports to the Editor). Please note this decision does not guarantee eventual acceptance.

Please submit a copy of your revised paper before 24-Apr-2019. Please note that the revision deadline will expire at 00.00am on this date. If we do not hear from you within this time then it will be assumed that the paper has been withdrawn. In exceptional circumstances, extensions may be possible if agreed with the Editorial Office in advance. We do not allow multiple rounds of revision so we urge you to make every effort to fully address all of the comments at this stage. If deemed necessary by the Editors, your manuscript will be sent back to one or more of the original reviewers for assessment. If the original reviewers are not available, we may invite new reviewers.

If your study uses humans or animals please include details of the ethical approval received, including the name of the committee that granted approval. For human studies please also detail

whether informed consent was obtained. For field studies on animals please include details of all permissions, licences and/or approvals granted to carry out the fieldwork.

- Data accessibility

If you wish to submit your supporting data or code to Dryad (<http://datadryad.org/>), or modify your current submission to dryad, please use the following link:
<http://datadryad.org/submit?journalID=RSOS&manu=RSOS-190174>

- Competing interests

- Authors' contributions

- Acknowledgements

- Funding statement

Kind regards,
Royal Society Open Science Editorial Office
Royal Society Open Science

on behalf of Professor Essi Viding
openscience@royalsociety.org

Editor's comments (Dr Essi Viding):

Two reviewers have taken time to go over the manuscript and provide helpful feedback. I welcome a revision of this manuscript, provided that the reviewer concerns are addressed. Both reviewers were enthusiastic about the manuscript and I hope you will be able to respond to their comments and submit a revised manuscript.

Comments to Author:

Reviewers' Comments to Author:

Reviewer: 1

Comments to the Author(s)

See in the attached

Reviewer: 2

Comments to the Author(s)

The goal of this manuscript was to test the associations among two different and commonly used measures of dietary restraint, the Restraint Scale and the Dutch Eating Behavior Questionnaire Restrained Eating (DEBQ-RE) in four different samples. Further, the authors tested the associations of each of these scales with body mass index, non-homeostatic eating (defined as external and disinhibited eating), and food craving (to chocolate and general foods), and compared the strength of the associations with each of the restraint scales with these external correlates. The authors also tested the factor structure and internal consistency of the Restraint Scale and DEBQ-RE. First, the authors found that the Restraint Scale and DEBQ-RE were significantly correlated. Second, dietary restraint was differentially associated with external and disinhibited eating across samples; however, overall, the Restraint Scale demonstrated larger associations with these non-homeostatic eating processes compared to the DEBQ-RE. In general, the RS demonstrated a stronger positive correlation to food craving and body mass index than the DEBQ-RE. Finally, the authors found that the DEBQ-RE and Restraint Scale factor structures replicated across samples.

Strengths of the study include a large sample size and the study helps the field move toward better understanding the multi-faceted nature of dietary restraint and differential associations among external correlates. The authors are also very clearly familiar with the dietary restraint literature. However, despite these strengths, I have reservations about the manuscript in its current form. My strongest concerns lie within the Introduction, as I feel that the Introduction lacks focus and could be significantly streamlined and also lacks scientific hypotheses to frame the research study. Other, more minor concerns are detailed in the review.

Overall Comments

Please be sure to use person-first language throughout the manuscript when describing weight status in persons. For example, it would be preferable to use the phrase "individuals with obesity" (vs. obese individuals).

This is very minor, but please spell out the first use of body mass index before using the abbreviation of BMI.

Introduction

The authors clearly are well-versed in the dietary restraint literature and the Introduction contains a lot of wonderful information about dietary restraint. At the same time, the Introduction feels a bit unfocused and quite long. I think the manuscript would be much improved if the Introduction were streamlined.

I also would love for the authors to enumerate hypotheses at the end of the Introduction to help frame the study. Based on the prior research, what did you think you might find?

In the first paragraph, I also disagree with the statement that "Overall, rigid diets are not only ineffective for weight loss..." Prescribed, medically monitored very-low calorie diets are effective in promoting weight loss in the short-term. The authors may want to revise this sentence to say that rigid diets are not effective in the long-term (or something of the like).

Method

I would like to see more evidence of reliability and validity of scale scores presented in the description of each measure. Moreover, I would like for internal consistency values to be separated and specified within the description of each measure.

Please report body mass index descriptive statistics (mean, standard deviation, range) for each sample in the Method section rather than in the Results section. BMI is a standard demographic variable for study such as this.

How were missing data handled? I am sure there were some missing data across the four different samples and large overall sample size.

What were the fit statistics of the reported factor analyses? Did the authors use parallel analysis to determine the number of factors to extract?

Were study measures completed online or in-person?

Results

Results were somewhat overwhelming in the way that they were presented. The authors might consider a more concise presentation of their results.

Discussion

Throughout the Discussion, the authors should tone down their conclusions given that this a cross-sectional study.

Author's Response to Decision Letter for (RSOS-190174.R0)

See Appendix B.

Decision letter (RSOS-190174.R1)

10-May-2019

Dear Dr Adams,

I am pleased to inform you that your manuscript entitled "Do restrained eaters show increased BMI, food craving, and disinhibition?

A comparison of the Restraint Scale and the DEBQ Restrained Eating scale" is now accepted for publication in Royal Society Open Science.

Kind regards,

Andrew Dunn

on behalf of Prof Essi Viding (Subject Editor)

Editor Comments to Author (Dr Essi Viding):

The authors have been very responsive, I think we can accept this now.

Follow Royal Society Publishing on Twitter: [@RSocPublishing](https://twitter.com/RSocPublishing)

Appendix A

Review

Title: Do restrained eaters show increased BMI, food craving, and disinhibited eating?
A comparison of the Restraint Scale and the Restrained Eating scale of the Dutch Eating Behaviour Questionnaire

Thank you very much for this really interesting paper, comparing to commonly used measures of dietary restraint, the DEBQ-Restaing subscale and the Restraint scale. I think the paper would be of great interest to eating behaviour researchers. See below for some comments.

Abstract

Thank you for the concise abstract. Please add the exact number of participants included, instead of “more than 1700 participants”. In addition, I would suggest adding the key results as well, such as the correlation coefficient between the scales, and the range of correlations coefficients between the two scales and the other phenotypes.

Introduction

Overall an interesting introduction covering a lot of the theoretical and research background. I believe, that at four pages, the introduction is too long, and some of the information especially in the end is either redundant or could be moved to other sections. The introduction is longer than the discussion, which seems a bit unusual. See below for more specific comments.

1. Page 3, from line 7: One suggestion would be to remove the section focussing on restraint and impulsivity. Impulsivity is not included in the later study, and I would propose to start with the mixed evidence around restraint and BMI.
2. Page 4, from line 3: I would recommend adding a reference for the statement that recommendations around restraint have been relaxed.
3. Page 4, line 17: Please rephrase “morbidly obese adults” using person-first language to avoid stigmatising language. One option would be: “adult with morbid obesity”.
4. Page 6: I get the impression that there is too much detail here, and some of the descriptors of the sample etc, should be moved to the methods section of the paper or deleted e.g. from line 8 to line 36.

Some information is repeated eg: line 52 “We analyse data from more than 1700 participants making it the largest study of its kind...”.

In addition, the final lines of the introduction seem to summarise the findings of the current study and I would suggest removing this. In my opinion a clear statement of the aims such as in line 46: “The aim of this study was to explore potential differences across restraint measures so that we could help to identify the most appropriate prescreening tool for eating-related interventions” is sufficient for the introduction and leave all the details about sample selection etc. for the methods.

Methods

Participants

1. I would suggest removing the information regarding the prescreening as it does not seem relevant.
2. The participant section could be restructured to something like: There were four samples included in this study. Sample 1 (N=xxx) consisted of staff and students from Cardiff university (x% female, age mean, range). Sample 2 (N=xx) included of undergrad psychology students (x% female, age mean, range). Sample 3 etc...

Overall, the total sample was XXX. Table 1 shows the different measures available in the different samples”

Measures and materials

Please add an example item for each of the mentioned scales so the readers unfamiliar with field can get an insight in what a typical question might be like.

Analyses

I would suggest breaking up the analyses section with subheading so it is easier to follow, as it was done in the results section: Internal consistency, Factor structure, Similarities and differences between scales, Statistical power

Results

Before giving more details of the results, a summary table of all data included would be useful including scores on scales, ages, %sex etc for the four samples.

Overall, the results sections sometimes veers into discussing the results, such as “Consistent with previous findings...” (page 11, line 36). I would suggest removing these and keeping them for the discussion.

As a reader, I would have preferred to have Table S.5 in the main manuscript instead of Table 2, as it shows all correlations across all measures, which I feel, might be seen as the main results and should be put to the forefront.

Discussion

I would suggest removing the first sentence and references and starting with “In the largest...”

Page 16, line: 26: I would be worried about over-interpreting the non-significant associations between DEBQRE and BMI and their direction. Please rephrase.

Page 17, last sentence: Maybe change “The Restraint Scale” in to RS for consistency.

Overall, I am missing critical evaluation of the study. What were the biggest strength and limitations of this work? For example, it seems that the ratio between men and women was unbalanced, as seen in Table S.4. The majority of the sample were women, which also rated higher on all the different scales. What effect would that have on the validity of the scales? Is one questionnaire maybe more relevant for men versus women? Just a few points to consider.

I hope these comments are helpful!

Thank you

Moritz Herle, PhD

Appendix B

We thank the reviewers for their expert and constructive appraisals of our manuscript. We respond below to each issue raised.

Reviewer 1

Abstract

Thank you for the concise abstract. Please add the exact number of participants included, instead of “more than 1700 participants”. In addition, I would suggest adding the key results as well, such as the correlation coefficient between the scales, and the range of correlations coefficients between the two scales and the other phenotypes.

The number of participants has been added to the abstract along with the key results as suggested.

Introduction

Overall an interesting introduction covering a lot of the theoretical and research background. I believe, that at four pages, the introduction is too long, and some of the information especially in the end is either redundant or could be moved to other sections. The introduction is longer than the discussion, which seems a bit unusual. See below for more specific comments.

We have followed your suggestions and the introduction has now been shortened by removing the paragraph on impulsivity and unnecessary methodological information.

1. Page 3, from line 7: One suggestion would be to remove the section focussing on restraint and impulsivity. Impulsivity is not included in the later study, and I would propose to start with the mixed evidence around restraint and BMI.

This section has now been removed

2. Page 4, from line 3: I would recommend adding a reference for the statement that recommendations around restraint have been relaxed.

A reference has now been added:

‘The association between dietary restraint and overeating has led to recommendations to relax restraint in order to promote healthy eating (Polivy & Herman, 1992); however, this has proved controversial due to inconsistencies across findings.’

3. Page 4, line 17: Please rephrase “morbidly obese adults” using person-first language to avoid stigmatising language. One option would be: “adult with morbid obesity”.

This has now been rephrased as “adults with morbid obesity”

4. Page 6: I get the impression that there is too much detail here, and some of the descriptors of the sample etc, should be moved to the methods section of the paper or deleted e.g. from line 8 to line 36.

Some information is repeated eg: line 52 “We analyse data from more than 1700 participants making it the largest study of its kind...”.

The final paragraph of the introduction has been shortened to only include relevant details. Other methodological information has been removed or moved to the method section as appropriate.

In addition, the final lines of the introduction seem to summarise the findings of the current study and I would suggest removing this. In my opinion a clear statement of the aims such as in line 46: "The aim of this study was to explore potential differences across restraint measures so that we could help to identify the most appropriate prescreening tool for eating-related interventions" is sufficient for the introduction and leave all the details about sample selection etc. for the methods.

We have taken the advice of reviewers 1 and 2 and have re-written the final section of the introduction as follows:

'The aim of this study was to explore potential differences across restraint measures so that we could help to identify the most appropriate prescreening tool for eating-related interventions. In accordance with previous research we expected to find that the RS was more strongly associated with disinhibited eating and BMI compared to the DEBQRE.'

Methods

Participants

1. I would suggest removing the information regarding the prescreening as it does not seem relevant.

This has been removed.

2. The participant section could be restructured to something like: There were four samples included in this study. Sample 1 (N=xxx) consisted of staff and students from Cardiff university (x% female, age mean, range). Sample 2 (N=xx) included of undergrad psychology students (x% female, age mean, range). Sample 3 etc... Overall, the total sample was XXX. Table 1 shows the different measures available in the different samples"

Thank you for your suggestion, this section has been simplified as detailed above. We have retained the information detailing that participants responded to an advert on food and positive emotion because we believe that this has implications for how the sample could potentially be biased (also now covered in the discussion).

Measures and materials

Please add an example item for each of the mentioned scales so the readers unfamiliar with field can get an insight in what a typical question might be like.

Example questions have been added to all questionnaires including for each subscale.

Analyses

I would suggest breaking up the analyses section with subheading so it is easier to follow, as it was done in the results section: Internal consistency, Factor structure, Similarities and differences between scales, Statistical power

This section has now been restructured with subheadings.

Results

Before giving more details of the results, a summary table of all data included would be useful including scores on scales, ages, %sex etc for the four samples.

A summary table of results has been added at the beginning of the results section.

Overall, the results sections sometimes veers into discussing the results, such as “Consistent with previous findings...” (page 11, line 36). I would suggest removing these and keeping them for the discussion.

Comments related to existing literature have been maintained for the internal consistency, factor analysis and demographic difference sections to demonstrate that our findings are comparable. These findings are not raised in the discussion - for clarity we have concentrated on the main aims/ findings of the study in the discussion. Other such comments have been moved from the results section into the discussion.

As a reader, I would have preferred to have Table S.5 in the main manuscript instead of Table 2, as it shows all correlations across all measures, which I feel, might be seen as the main results and should be put to the forefront.

Thank you for this suggestion, this table has now been added to the main manuscript.

Discussion

I would suggest removing the first sentence and references and starting with “In the largest...”

The first paragraph of the discussion has been rewritten:

‘In the largest study of its kind, we reveal that although the RS and DEBQRE are highly correlated with one another, there are also significant differences between the two measures. In support of existing literature we demonstrate that the RS is more strongly associated with external and disinhibited eating, food craving and BMI, when compared to the DEBQRE...’

Page 16, line: 26: I would be worried about over-interpreting the non-significant associations between DEBQRE and BMI and their direction. Please rephrase.

This has now been rephrased:

‘Associations with the DEBQRE did not reach statistical significance.’

Page 17, last sentence: Maybe change “The Restraint Scale” in to RS for consistency.

This has been changed.

Overall, I am missing critical evaluation of the study. What were the biggest strength and limitations of this work? For example, it seems that the ratio between men and women was unbalanced, as seen in Table S.4. The majority of the sample were women, which also rated higher on all the different scales. What effect would that have on the validity of the scales? Is one questionnaire maybe more relevant for men versus women? Just a few points to consider.

We have now included a paragraph in the discussion on limitations and future directions.

Reviewer: 2

My strongest concerns lie within the Introduction, as I feel that the Introduction lacks focus and could be significantly streamlined and also lacks scientific hypotheses to frame the research study. Other, more minor concerns are detailed in the review.

Responses to these comments are addressed below.

Overall Comments

Please be sure to use person-first language throughout the manuscript when describing weight status in persons. For example, it would be preferable to use the phrase “individuals with obesity” (vs. obese individuals).

This has now been rephrased as “adults with morbid obesity”

This is very minor, but please spell out the first use of body mass index before using the abbreviation of BMI.

This has been done in the abstract and the first instance in the main manuscript.

Introduction

The authors clearly are well-versed in the dietary restraint literature and the Introduction contains a lot of wonderful information about dietary restraint. At the same time, the Introduction feels a bit unfocused and quite long. I think the manuscript would be much improved if the Introduction were streamlined.

The introduction has now been reduced in length. We have removed the second paragraph and condensed the final two paragraphs consistent with reviewer 1’s comments.

I also would love for the authors to enumerate hypotheses at the end of the Introduction to help frame the study. Based on the prior research, what did you think you might find?

We have added a hypothesis to the end of the introduction regarding disinhibited eating and BMI as these variables were supported by the literature:
‘In accordance with previous research we expected to find that the RS was more strongly associated with disinhibited eating and BMI compared to the DEBQRE.’

We have not added hypotheses regarding chocolate craving or external eating because these were exploratory questions which we did not have a priori hypotheses for.

In the first paragraph, I also disagree with the statement that “Overall, rigid diets are not only ineffective for weight loss...” Prescribed, medically monitored very-low calorie diets are effective in promoting weight loss in the short-term. The authors may want to revise this sentence to say that rigid diets are not effective in the long-term (or something of the like).

We cannot find any comments on rigid diets within the manuscript.

Method

I would like to see more evidence of reliability and validity of scale scores presented in the description of each measure. Moreover, I would like for internal consistency values to be separated and specified within the description of each measure.

Measures of internal consistency and test-retest reliability have been added to the measures where available. We have also referred the reader to discussions on validity for the RS, which are beyond the scope of this paper.

Please report body mass index descriptive statistics (mean, standard deviation, range) for each sample in the Method section rather than in the Results section. BMI is a standard demographic variable for study such as this.

We only have BMI statistics for two of the samples included (one self-report and one measured) and have therefore not included BMI statistics in the participants section. In line with reviewer 1's comments we have added them to a summary table of descriptive statistics at the beginning of the results.

How were missing data handled? I am sure there were some missing data across the four different samples and large overall sample size.

The treatment of missing data has been added to the statistical analysis section: 'To account for missing data across questionnaires, all correlations use individual mean scores rather than the total scores. Cases were removed from analyses where no data was available.'

What were the fit statistics of the reported factor analyses? Did the authors use parallel analysis to determine the number of factors to extract?

Details of the factor analysis have been moved to the statistical analysis section to make these more apparent and KMO values are reported in the supplementary information: 'The factor structure of the RS and DEBQRE were explored using principle components analysis with varimax rotation in accordance with previous research (Allison et al., 1992; Blanchard & Frost, 1983; Laessle et al., 1989; Ruderman, 1983). Factors were extracted based on having eigenvalues greater than 1.'

Were study measures completed online or in-person?

This information has now been added to the procedure: 'All participants completed the questionnaires remotely. Those in samples 2 and 3 completed the questionnaires via an internet survey whereas those in sample 1 could choose to answer the questionnaires electronically via email, in hard copy or via an internet survey.'

Results

Results were somewhat overwhelming in the way that they were presented. The authors might consider a more concise presentation of their results.

We have now included a summary table of all results to help the reader (Table 3). We have also removed / moved elements to help streamline the results where possible.

Discussion

Throughout the Discussion, the authors should tone down their conclusions given that this a cross-sectional study.

We have ensured that cautious and associative language is used throughout the discussion, consistent with the recommendations of Adams et al. (2017) for how to report observational studies.